# Nimodipine Exerts Beneficial Effects on the Rat Oligodendrocyte Cell Line OLN-93

**DOI:** 10.3390/brainsci12040476

**Published:** 2022-04-04

**Authors:** Felix Boltz, Michael Enders, Andreas Feigenspan, Philipp Kirchner, Arif Ekici, Stefanie Kuerten

**Affiliations:** 1Institute of Anatomy and Cell Biology, Friedrich-Alexander University of Erlangen-Nuremberg, 91054 Erlangen, Germany; felix.boltz@tutanota.com (F.B.); michael.enders@uni-bonn.de (M.E.); 2Institute of Neuroanatomy, Medical Faculty, University of Bonn, 53115 Bonn, Germany; 3Department of Biology, Animal Physiology, Friedrich-Alexander University of Erlangen-Nuremberg, 91058 Erlangen, Germany; andreas.feigenspan@fau.de; 4Institute of Human Genetics, University Hospital Erlangen, Friedrich-Alexander University of Erlangen-Nuremberg, 91054 Erlangen, Germany; philipp.kirchner@fau.de (P.K.); arif.ekici@uk-erlangen.de (A.E.)

**Keywords:** dihydropyridines, MS, myelination, neuroprotection, nimodipine, OLN-93

## Abstract

Multiple sclerosis (MS) is a chronic autoimmune disease of the central nervous system (CNS). Therapy is currently limited to drugs that interfere with the immune system; treatment options that primarily mediate neuroprotection and prevent neurodegeneration are not available. Here, we studied the effects of nimodipine on the rat cell line OLN-93, which resembles young mature oligodendrocytes. Nimodipine is a dihydropyridine that blocks the voltage-gated L-type calcium channel family members Ca_v_1.2 and Ca_v_1.3. Our data show that the treatment of OLN-93 cells with nimodipine induced the upregulation of myelin genes, in particular of proteolipid protein 1 (*Plp1*), which was confirmed by a significantly greater expression of PLP1 in immunofluorescence analysis and the presence of myelin structures in the cytoplasm at the ultrastructural level. Whole-genome RNA sequencing additionally revealed the upregulation of genes that are involved in neuroprotection, remyelination, and antioxidation pathways. Interestingly, the observed effects were independent of Ca_v_1.2 and Ca_v_1.3 because OLN-93 cells do not express these channels, and there was no measurable response pattern in patch-clamp analysis. Taking into consideration previous studies that demonstrated a beneficial effect of nimodipine on microglia, our data support the notion that nimodipine is an interesting drug candidate for the treatment of MS and other demyelinating diseases.

## 1. Introduction

Multiple sclerosis (MS) is a chronic autoimmune disease of the central nervous system (CNS). With over 2 million patients affected worldwide, MS is the most common neurological disorder in young adults [1]. The disease leads to irreversible functional deficits, the requirement for long-term medication, and premature retirement, therefore causing a significant socioeconomic burden [2]. Although the exact causes of MS are unknown, a multifactorial aetiology is assumed, including genetic predisposition, viral and bacterial infections, dietary factors, and vitamin D deficiency [3,4,5,6]. MS is characterised by an immune response against antigens of the CNS, including oligodendrocyte/myelin, neuronal, and axonal antigens [7,8]. The clinical course of MS can be classified into relapsing–remitting MS (RRMS), which is characterised by clearly defined attacks of new or increasing neurologic symptoms followed by periods of partial or complete recovery (remissions), and secondary progressive MS (SPMS) that follows an initial relapsing–remitting course and shows progressive worsening of neurologic function over time. Furthermore, there is primary progressive MS (PPMS) that is characterised by worsening neurologic function from the onset of symptoms. In the progressive forms of MS, occasional relapse episodes of intensified symptoms similar to those experienced by RRMS may occur [7]. While the disease is incurable, there are several therapeutic options available, predominantly based on the concept of immune modulation and suppression, which slow down disease progression and attenuate clinical symptoms [9]. In addition to inflammation, neurodegeneration within the CNS plays an important role in the immunopathology of MS, particularly in the progressive stages [10,11].

There are ongoing efforts to develop drugs that are capable of promoting CNS neuroprotection and repair in patients with MS, but so far, none of the potential drug candidates has been approved for clinical practice [12]. Most recently, a phase III clinical trial in patients with progressive MS using high-dose biotin treatment did not meet its primary endpoint [13]. The monoclonal antibody opicinumab, which targets the transmembrane signalling protein leucine-rich repeat and immunoglobin-like domain-containing nogo receptor-interacting protein 1 (LINGO1), considered to be a key negative regulator of myelination [14], did not significantly improve clinical symptoms in patients with relapsing MS and acute optic nerve neuritis [15,16].

Nimodipine belongs to a group of dihydropyridines and is a prototypical voltage-gated L-type calcium channel blocker that acts on Ca_v_1.1–Ca_v_1.4 [17,18]. Since the drug is hydrophobic in nature, it can cross the blood–brain barrier. Nimodipine was approved by the FDA in 1988 for the prevention and treatment of neurologic deficits in patients with aneurysmal subarachnoid haemorrhage, and the main mode of action that was initially proposed was the relaxation of cerebral vascular smooth muscle [17,19,20].

Since then, several preclinical studies have demonstrated additional beneficial effects of nimodipine on the CNS, including the prevention of cognitive dysfunction after surgery [21] and an increase in cognitive function after cerebral ischaemia in rats [22]. In addition, nimodipine improved symptoms in experimental autoimmune encephalomyelitis (EAE), a common mouse model of MS, fostering remyelination, reducing demyelination, and increasing the number of oligodendrocyte transcription factor (OLIG)2- and adenomatous polyposis coli (APC)-positive oligodendrocytes [23,24]. Improved remyelination was also observed in nimodipine-treated mice, in which demyelination was induced with the copper chelator cuprizone [25]. This study suggested an astrocyte-mediated effect of nimodipine on myelination. Clinical studies showed a positive effect on vocal fold and facial motion recovery in patients after recurrent laryngeal and facial nerve injury [26], as well as an improved hearing outcome after vestibular schwannoma surgery [27].

Overall, nimodipine seems to affect several different cell types of the nervous system, including oligodendrocytes, Schwann cells, astrocytes, microglia, and neurons, in part via a Ca_v_1.2-independent mechanism [23,28]. The current study aimed to address the question of whether nimodipine exerts direct effects on oligodendrocytes using the rat oligodendrocyte cell line OLN-93.

## 2. Materials and Methods

### 2.1. Cells and Cell Culture

The rat oligodendrocyte cell line OLN-93 was provided by Christiane Richter-Landsberg (University of Oldenburg, Oldenburg, Germany). Cells were grown in T75 cell culture flasks (Thermo Fisher Scientific, Waltham, MA, USA) using Dulbecco’s modified Eagle’s medium (DMEM; Thermo Fisher Scientific) supplemented with 10% (*v*/*v*) foetal bovine serum (FBS) (Thermo Fisher Scientific) and 1% penicillin/streptomycin (Thermo Fisher Scientific) (93-medium). Cells were maintained at 37 °C and 5% CO_2_ and passaged when confluency of 80–90% was reached. All cell culture materials were coated overnight with 0.0075 mg/mL poly-D-lysine (PDL; Sigma-Aldrich, St. Louis, MO, USA) before use. For the collection of cells, the medium was removed, cells were washed with phosphate-buffered saline (PBS; Thermo Fisher Scientific) and incubated with 0.1% trypsin (Thermo Fisher Scientific) in PBS under microscopic control for 5–8 min until cells detached from the cell culture flask. The process was stopped by adding 93-medium. Cells were centrifuged at room temperature at 300× *g* for 5 min and further processed or passaged into a new cell culture flask.

### 2.2. Nimodipine

Nimodipine was provided at research grade (batch: BXR4H3P) by Bayer AG (Leverkusen, Germany) following a material transfer agreement and dissolved in dimethyl sulfoxide (DMSO; Thermo Fisher Scientific) to obtain a 10 mM stock concentration.

### 2.3. Reverse Transcription Polymerase Chain Reactions

For examination of Ca_v_1.2 and Ca_v_1.3 expression by OLN-93 cells, cells were grown in 93-medium for several days at 37 °C and 5% CO_2_ until confluency of 80–90% was reached. They were then transferred into OLN-93 differentiation medium (93D-medium), which consisted of DMEM containing 0.5% (*v*/*v*) FBS and 1% (*v*/*v*) penicillin/streptomycin. Cells were harvested on culture Days 0, 1, 2, 4, and 6. Total RNA was isolated using the RNeasy Plus Mini Kit (Qiagen, Hilden, Germany), following the manufacturer’s instructions. Reverse transcription was performed using the High-Capacity cDNA Reverse Transcription Kit (Thermo Fisher Scientific). A polymerase chain reaction was performed with the PCR Red MasterMix (Genaxxon Bioscience GmbH, Ulm, Germany) using the ProFlex PCR Systems Cycler (Thermo Fisher Scientific). Primers were designed using the National Center for Biotechnology Information Primer Blast (https://www.ncbi.nlm.nih.gov/tools/primer-blast/ accessed on 1 February 2020) and purchased from Thermo Fisher Scientific (USA). Primer sequences are shown in Table 1. Gel electrophoresis was performed for 1.5 h at 100 V using a freshly prepared 2% agarose gel (Genaxxon) with GelRed DNA staining dye (Genaxxon). GenLadder 100 bp plus 1.5 kbp (Genaxxon) was used as the molecular weight marker. Results were documented using an iBright CL1500 imaging system (Thermo Fisher Scientific).

### 2.4. Immunocytochemistry

In preparation for immunocytochemical staining, 12 mm-diameter coverslips (Carl Roth GmbH & Co. KG, Karlsruhe, Germany) were sterilised in 1 M HCl overnight, washed with double distilled water (ddH_2_O), placed in 70% ethanol for 4 h, and UV sterilised at room temperature for 30 min. For the characterisation of OLN-93 cells, 2–3 × 10^4^ cells were seeded per well of a 24-well plate containing one coverslip each and incubated in 93-medium for 24–48 h. For proteolipid protein (PLP)1 immunofluorescence quantification, 2–3 × 10^4^ cells were seeded as above and incubated in 93D-medium that contained either 10 µM nimodipine or an equal amount of DMSO as vehicle control. Cells were analysed after 6 days, following which the medium was removed; the cells were washed with PBS and fixed in ice-cold 4% (*v*/*v*) paraformaldehyde (Carl Roth) in PBS for 10 min. After the removal of paraformaldehyde, cells were washed three times with cold PBS and stored in PBS at 4 °C. For staining, cells were permeabilised using 0.1 M triton X (Carl Roth) at room temperature for 10 min, washed three times with Tris-buffered saline (TBS) and blocked with 2% (*w*/*v*) bovine serum albumin (Carl Roth) in TBS containing 5% (*v*/*v*) goat serum (Sigma-Aldrich) for 1 h at room temperature. After blocking, the primary antibodies for staining myelin basic protein (MBP) (dilution 1:1000 in blocking solution; Abcam, Cambridge, UK [29]), PLP1 (dilution 1:500 in blocking solution; Abcam [30]), and and sex-determining region Y (SRY)-related high mobility group-box (SOX)10 (dilution 1:500 in blocking solution; Abcam [31]) were prepared in blocking solution, and the cells were incubated at 4 °C overnight. The cells were then washed with PBS and incubated with the secondary antibodies goat anti-chicken Cy3 (dilution 1:500 in blocking solution; Dianova, Hamburg, Germany [32]) or goat anti-rabbit AlexaFluor^TM^ 488 (dilution 1:1000 in blocking solution; Molecular Probes, Eugene, OR, USA [33]) at room temperature for 1 h. Coverslips were washed with PBS and ddH_2_O and mounted in Fluoroshield mounting medium with DAPI (Abcam). Images were acquired using a Leica DM5 B fluorescence microscope (Leica, Wetzlar, Germany). For analysis of the PLP1 fluorescence signal, the concept of corrected total CTCF was applied, as previously described [34], where:CTCF *=* Integrated density − (Area selected × Mean fluorescence of background readings)

The mean of three independent background readings was used for calculations. The CTCF was quantified in relation to the DAPI-positive area (CTCF*_DAPI_*) using a custom-made ImageJ macro. Three independent sample runs were performed, and 174–438 cells were measured per sample run.

### 2.5. Patch-Clamp Analysis

Patch-clamp recordings were carried out as described previously [35]. Briefly, coverslips with adherent oligodendrocytes were transferred to the recording chamber of an upright microscope (Zeiss, Jena, Germany) with a solution containing: 137 mM NaCl, 5.4 mM KCl, 1.8 mM CaCl_2_, 1 mM MgCl_2_, 5 mM (4-(2-hydroxyethyl)-1-piperazineethanesulfonic acid (HEPES), and 10 mM glucose (pH 7.4). The patch pipette solution contained: 120 mM K-D-gluconate, 20 mM KCl, 1 mM CaCl_2_, 2 mM MgCl_2_, 11 mM ethylene glycol tetraacetic acid (EGTA), and 10 mM HEPES (pH 7.2). All experiments were carried out at room temperature (22–23 °C). Voltage-gated currents were recorded with an EPC10 patch-clamp amplifier (Heka Elektronik, Lambrecht, Germany), low-pass filtered at 2.9 kHz using a built-in Bessel filter, and digitised at 20 kHz with Patchmaster software (Heka Elektronik). Patch electrodes were pulled from borosilicate glass (Hilgenberg, Malsfeld, Germany) to a final resistance of 3–5 MΩ. Electrode tips were coated with Sylgard 184 (Dow Corning, Midland, MI, USA), and their series resistance (5–10 MΩ) was compensated up to 80%. Current traces were analysed and plotted with OriginPro (OriginLab Corporation, Northampton, MA, USA) and MATLAB (Mathworks, Natick, MA, USA).

### 2.6. Real-Time Quantitative Polymerase Chain Reaction

For RT-qPCR, 5 × 10^5^ cells were cultured in PDL-coated T75 cell culture flasks containing 93-medium and incubated overnight. On the next day (Day 0), the medium was removed, cells were washed with PBS, and 93D-medium containing 10 µM nimodipine or an equal volume of DMSO as vehicle control was added. Cells were processed on Day 0 and Day 6. Day 0 (before the addition of the nimodipine/vehicle control) was used for baseline measurements. Total RNA was extracted using the PureLink RNA Mini Kit (Thermo Fisher Scientific), including DNase treatment (Thermo Fisher Scientific) according to the manufacturer’s instructions. RNA concentration and purity were measured using the Eppendorf BioPhotometer Plus (Eppendorf, Hamburg, Germany). Reverse transcription was performed using the High-Capacity cDNA Reverse Transcription Kit (Thermo Fisher Scientific). RT-qPCR was performed using the StepOnePlus Real-Time PCR System (Thermo Fisher Scientific) and TaqMan Fast Advanced Mastermix (Thermo Fisher Scientific). Relative gene expression was assessed using TaqMan assays (Thermo Fisher Scientific) for rat *Mbp* (Assay ID: Rn01399619_m1), rat *Plp1* (Assay ID: Rn01410490_m1), and rat *Actb* (Assay ID: Rn00667869_m1) genes, the latter as an endogenous control. Relative gene expression was assessed using the ΔΔCT method, as previously described [36].

### 2.7. Transmission Electron Microscopy

A total of 5 × 10^5^ OLN-93 cells were cultured overnight in 93-medium. On the next day, the medium was removed, cells were washed with PBS and 93D-medium that contained 10 µM nimodipine or an equal volume of DMSO as vehicle control was added to the cells. Cells were incubated for 6 days. Cells were washed twice with PBS and incubated with 2.5% (*v*/*v*) glutardialdehyde (Carl Roth) in PBS at 4 °C for 2 h. Cells were then washed twice with PBS and stored in PBS at 4 °C overnight. Cells were treated with 3% (*w*/*v*) potassium ferricyanide (Merck, Darmstadt, Germany) and 1% (*w*/*v*) osmium tetroxide (Emsdiasum, Hatfield, PA, USA) in PBS at room temperature for 2 h and incubated in 0.1 M phosphate buffer (Sigma-Aldrich) at 4 °C overnight. Samples were embedded in 2% (*w*/*v*) agar (Merck) in 0.1 M phosphate buffer (Sigma-Aldrich) before they were treated with 70% ethanol (Carl Roth) for 60 min, followed sequentially by 80%, 90%, and 100% ethanol, 100% ethanol (undenatured), 100% ethanol/acetone (Carl Roth) 1/1, acetone only, 2/3 acetone and 1/3 Epon, 1/3 acetone, and 2/3 Epon for 30 min each, and finally Epon and 2% glycidether accelerator DMP-30 (Carl Roth) for 180 min. Epon was prepared by mixing solution A, consisting of 75 mL glycidether 100 (Carl Roth) and 120 mL of glycidether hardener dibenzylideneacetone (Carl Roth), with solution B, consisting of 120 mL of glycidether 100 and 105 mL of glycidether hardener methyl nadic anhydride (Carl Roth). After dehydration, samples were aligned in moulds, coated with Epon and 2% glycidether accelerator DMP-30 and polymerised at 60 °C and 80 °C overnight. Epon blocks were cut into 50–60 nm-thick sections, stretched with chloroform (Carl Roth) and transferred to copper grids (Science Services, Munich, Germany). Samples were contrasted for 10 min using 10% (*w*/*v*) uranyl acetate (Emsdiasum) and rinsed with ddH_2_O. Finally, samples were treated with 2.8% (*w*/*v*) lead citrate (Merck) for 10 min. Image acquisition was performed using a Zeiss EM 906 transmission electron microscope (Zeiss) with a cathode voltage of 60 kV and the TRS 2048 digital camera system (Tröndle Restlicht Verstärkersysteme, Moorenweis, Germany). For each sample, approximately 50 random images at 6000× magnification were used to count the number of cells containing clearly visible myelin formations. The experiment was performed three times, and the proportion of cells with myelin formation was expressed as the percentage of the total number of counted cells.

### 2.8. Whole Transcriptome Analysis/RNA Sequencing

Cells were cultured and treated with nimodipine or a vehicle control as above and processed for sequencing on Day 1 and Day 6 of treatment. Total RNA was extracted using the PureLink RNA Mini Kit (Thermo Fisher Scientific), including DNase treatment. The quality of the isolated RNA samples was determined using an Agilent 2100 Bioanalyzer equipped with an Agilent RNA 6000 Nano kit and related software (Agilent, Santa Clara, CA, USA). Sequencing libraries were generated from 1 mg of high-quality RNA using the TruSeq Stranded mRNA Kit (Illumina, San Diego, CA, USA) according to the manufacturer’s instructions. Libraries were sequenced at a single end on a HiSeq 2500 platform (Illumina) with 100 bp reads to a depth of at least 30 million reads. Reads were converted to a FASTQ format with bcl2fastq (version 2.17.1.4; Illumina). Sequences matching Illumina adapters were masked using Cutadapt (version 1.18; https://doi.org/10.14806/ej.17.1.200 accessed on 8 February 2021), and reads with more than 40 masked bases were discarded. Before and after adapter masking, read quality was checked in Fastqc (version 0.11.8; Illumina). The remaining reads were mapped to the *Rattus norvegicus* reference genome Rnor 6.0, Ensembl gene annotation 101, using spliced transcripts alignment to a reference (STAR) software (version 2.6.1c; https://github.com/alexdobin/STAR accessed on 8 February 2021) and quantified as reads per gene while excluding exons shared between more than one gene (Subread; version 1.6.1; http://subread.sourceforge.net/ accessed on 8 February 2021). Mapping quality was investigated using RNA-SeQC (version 2.3.4; https://github.com/getzlab/rnaseqc/releases/tag/v2.3.4 accessed on 8 February 2021) and sequencing saturation was assessed by the rarefication of reads. Based on the read count per gene, differentially expressed genes were determined using the negative binomial model as implemented in the differential gene expression analysis package DESeq2 (version 1.28.1; http://www.bioconductor.org/packages/release/bioc/html/DESeq2.html; R version 4.0.2 [37] accessed on 8 February 2021). Results from significance tests were corrected for multiple testing using the Benjamini–Hochberg method [38]. Significantly differentially expressed genes in the nimodipine-treated samples compared with the vehicle on Day 1 and Day 6 were identified by a log2 fold change of ≤−0.4 and ≥0.4 in the context of oligodendrocytes, myelination, and MS. 

### 2.9. Statistical Analysis

Statistical analysis was performed using GraphPad Prism 9 (San Diego, CA, USA), except for the analysis of RNA sequencing data sets (for details, please see Section 2.8). Statistical significance was set to *p* < 0.05 and determined by Students *t*-test for parametric data sets or Mann–Whitney *U* test for non-parametric data sets.

## 3. Results

### 3.1. OLN-93 Cells Resemble Oligodendrocytes and do Not Express Ca_v_1.2 and Ca_v_1.3

To characterise the OLN-93 cell line under our laboratory conditions, we performed immunofluorescence staining using the oligodendrocyte markers MBP, PLP1, and SOX10. MBP and PLP1 are myelin antigens, which are known to be expressed by the cell line [39]; SOX10 is a transcription factor expressed by oligodendrocytes [40,41]. As shown in Figure 1A, OLN-93 cells were positive for all three markers. We then induced further differentiation of the cell line by culturing them in 93D-medium, which contained 0.5% (*v*/*v*) FBS compared with 10% (*v*/*v*) in the normal growth medium (93-medium)]. Light microscopic analysis showed morphological changes, including an increased arborisation of the cells under low FBS conditions (Figure 1B).

As the aim of the study was to investigate the effects of the L-type calcium channel antagonist nimodipine on OLN-93 cells, the RNA expression of Ca_v_1.2 and Ca_v_1.3 during the process of differentiation was also determined. While nimodipine blocks Ca_v_1.1–Ca_v_1.4, Ca_v_1.2 and Ca_v_1.3 are known to be expressed in the brain [42,43], and Ca_v_1.4 is predominantly expressed in the photoreceptor terminals of retinal neurons [44]. As shown in Figure 1C, the RNA expression of both Ca_v_1.2 and Ca_v_1.3 was not detectable in OLN-93 cells. We subsequently performed patch-clamp analysis (Figure 1D). Voltage-gated calcium channels were routinely activated by depolarising the membrane potential, resulting in an influx of Ca^2+^ ions. We carried out voltage-clamp recordings from cultured OLN-93 cells in the whole-cell configuration of the patch-clamp technique at a holding potential of −70 mV. Neither a single voltage step to 0 mV nor a series of increasing voltage steps induced a measurable current (*n* = 5) (Figure 1D).

### 3.2. Nimodipine Increases the Expression of Plp1 in OLN-93 Cells

We next examined the effects of nimodipine on the expression of the myelin genes *Plp1* and *Mbp*. Table 2 shows the results of the relative gene expression compared with Day 0 (baseline). On Day 6, *Plp1* expression showed a significant increase relative to the baseline in cells treated with 10 µM nimodipine (mean ± standard error [SE], 33.51 ± 12.91 in nimodipine-treated cells vs. 18.95 ± 7.631 in vehicle-treated cells, *n* = 6, *p* = 0.0028). In contrast, the effect of nimodipine on *Mbp* expression on Day 6 was limited (mean ± SE, 4.818 ± 0.7941 in nimodipine-treated cells vs. 4.026 ± 0.6479 in vehicle-treated cells, *n* = 6, *p* = 0.1081). These results encouraged us to further investigate the effects of nimodipine on myelin genes in OLN-93 cells.

### 3.3. Nimodipine Increases the Formation of Myelin at the Ultrastructural Level

Myelin-like structures could be identified in OLN-93 cells using transmission electron microscopy (Figure 2A,B). The percentage of cells with clearly visible myelin formation (Figure 2C) was significantly higher in cells that were treated with 10 µM nimodipine for 6 days compared with the vehicle (mean ± SE, 58.7% ± 11.7% vs. 25.4% ± 8.5%, respectively; *p* < 0.05, *n* = 3). To further investigate the effects of nimodipine at the protein level using an alternative method, the PLP1 fluorescence signal was quantified by setting the corrected total cellular fluorescence (CTCF) in relation to the 4′,6-diamidino-2-phenylindole (DAPI) area of the cells (CTCF*_DAPI_*). Representative images of the PLP1 staining are shown in Figure 3A. In all three independent runs, the CTCF*_DAPI_*was significantly higher in cells that were incubated with nimodipine compared with the vehicle solution (*p* < 0.01; Figure 3B).

### 3.4. Nimodipine Shifts RNA Expression towards Myelination and Maturation

As we were interested in exploring potential pathways that were targeted by nimodipine, we performed whole transcriptome analysis (RNA sequencing). The expression of common myelin-associated genes on Days 1 and 6 is presented in Figure 4A. Figure 4B displays the significantly differentially expressed genes that might be interesting candidates in the context of myelination of oligodendrocytes and MS. Typical myelin-associated genes, such as *Plp1* and peripheral myelin protein 2 (*Pmp2*) [45,46], were upregulated in nimodipine-treated cells. In addition, the early myelin marker cyclic nucleotide phosphodiesterase (*Cnp*) [47] was slightly upregulated, along with the oligodendrocyte differentiation markers *Olig1*, *Sox8,* and *Sox10* [41,48,49] on Day 1, and tenascin R (*Tnr*) and leukemia inhibitory factor (*Lif*) on Day 6, the latter being regulators of oligodendrocyte differentiation [50,51,52]. G protein-coupled receptor 37 (*Gpr37*) as well as tenascin C (*Tnc*), both of which are negative regulators of oligodendrocyte differentiation [51,53], were downregulated in nimodipine-treated cells. Interestingly, other myelin-associated genes, such as myelin-associated glycoprotein (*Mag*), myelin oligodendrocyte glycoprotein (*Mog*), and *Mbp* were not differentially expressed when comparing nimodipine- to vehicle-treated cells. Overall, these data suggest that nimodipine shifts RNA expression towards a promyelinating gene pattern.

Further genes involved in the regulation of (re)myelination, including guanine nucleotide exchange factor 3 (*Vav3*), *Lif,* heme oxygenase 1 (*Hmox1*)*,* Kruppel-like factor 9 (*Klf9*), and crystallin alpha B *(Cryab*) [52,54,55,56,57,58] were also upregulated by nimodipine (Figure 4B). Moreover, genes involved in cholesterol synthesis, such as very-low-density lipoprotein receptor (*Vldlr*), which were reported to be expressed in mature myelinating oligodendrocytes in the postnatal CNS of mice [59], were upregulated in nimodipine-treated cells (Figure 4B).

Along these lines, genes involved in Schwann cell myelination and associated with cholesterol synthesis, such as MAF bZIP transcription factor (*Maf*), neuregulin 1 (*Nrg1*), and ERBB2 receptor tyrosine kinase 3 (*Erbb3),* were also upregulated in nimodipine-treated cells [60]. In addition, fibulin 5 (*Fbln5*)*,* which is an important gene for Schwann cell development and peripheral nervous system remyelination [61], was upregulated.

Detoxification-associated genes, such as NAD(P)H quinone oxidoreductase 1 (*Nqo1*) and glutathione S-transferase alpha 1 (*Gsta1*) [62,63], showed higher expression in nimodipine-treated cells, as did the neuroprotection-associated gene glycoprotein NMB (*Gpnmb*) [64,65]. Additionally, we found altered expression of a subset of genes that was implicated in the *Notch1* signalling pathway, including glutathione-specific gamma-glutamylcyclotransferase 1 (*Chac1*), delta-like non-canonical Notch ligand 2 (*Dlk2*), and delta-like canonical Notch ligand 1 (*Dll1)*, potentially indicating *Notch1* pathway-associated neuroprotective and promyelinating effects. Overall, nimodipine seemed to shift the pattern of RNA expression towards neuroprotection, myelination and maturation of oligodendrocytes. A detailed analysis of the differentially expressed genes and their importance for myelination, differentiation, and cell protection in oligodendrocytes shown in Figure 4B is presented in Table 3.

## 4. Discussion

In a previous study, we showed that nimodipine has beneficial effects on neuroinflammation in EAE, a mouse model of MS [23]. Our data suggested that nimodipine inhibits the production of nitric oxide and reactive oxygen species by microglia and induces the apoptosis of proinflammatory microglia. Such effects on microglia might indirectly promote (re)myelination by creating a favourable anti-inflammatory environment. In support of this theory, we observed an increased number of OLIG2-positive and OLIG2/APC double-positive oligodendrocytes in the spinal cord of nimodipine-treated EAE mice [23]. In line with these findings, results from other studies suggested positive effects of nimodipine on postoperative cognitive dysfunction and cerebral ischaemia in animal models [21,22], in addition to a beneficial effect on human vocal fold and facial motion recovery [26] and hearing outcome after vestibular schwannoma surgery [27].

The aim of this study was to investigate whether nimodipine exerts direct effects on oligodendrocytes using the rat oligodendrocyte cell line OLN-93, which resembles premature, 5- to 10-day-old (postnatal time) oligodendrocytes [39]. Our data demonstrate increased *Plp1*/PLP1 expression both at the RNA and protein level. Interestingly, *Mbp* expression remained unaffected, suggesting specific and differential effects of the drug on myelin genes. This notion was supported by the gene patterns that were observed in RNA sequencing experiments of OLN-93 cells after incubation with nimodipine. Here, the data did not only imply a positive effect of nimodipine on selected myelin-associated genes but also on genes that play a role in antioxidant pathways and in the context of neuroprotection (Table 3). Interesting gene candidates belonged, e.g., to the *Notch* signalling pathway. The effect of nimodipine on OLN-93 cells seemed to be time-dependent because gene patterns differed on Day 1 and Day 6 of culture. Furthermore, we observed the intracellular formation of myelin structures on the ultrastructural level in OLN-93 cells following incubation with nimodipine.

Nevertheless, taking all results into consideration, OLN-93 cells did not show the typical pattern that would be expected from pre-oligodendrocytes differentiating into mature myelinating oligodendrocytes. Hence, the interpretation of a potential promyelinating effect of nimodipine needs to be taken with caution. In addition, the current study did not use a co-culture system of oligodendrocytes and neurons, which would be better suited to study the myelinating capacity of oligodendrocytes.

The OLN-93 cell line was previously used to investigate various biological processes in oligodendrocytes, including their response to oxidative stress [94,95], the neuroprotective effects of β-caryophyllene against lipopolysaccharide-induced oligodendrocyte toxicity [96], and ferroptosis in oligodendrocytes [97]. The effects of several drugs on oligodendrocytes, including fingolimod [98], apigenin [99], haloperidol, or clozapine [100], were also assessed using the OLN-93 cell line. Yet, a cell line does not entirely recapitulate primary cells or in vivo models. As OLN-93 cells do not resemble oligodendrocyte precursor cells (OPCs), we were only able to study the effects of nimodipine in the later stages of oligodendrogenesis. However, it would be interesting to determine if enhanced maturation was also observable in OPCs after nimodipine treatment. OPCs express Ca_v_1.2, and calcium influx through this channel was reported to be crucial for normal OPC development, survival, and myelination [101,102,103].

What remains unclear at this point is how nimodipine triggers the observed effects in oligodendrocytes. As with microglia, our data indicate that the effect on oligodendrocytes may be independent of Ca_v_1.2 and Ca_v_1.3 channels, as both were undetectable in OLN-93 cells by RT-qPCR and there was no measurable calcium influx in patch-clamp analysis. The results suggest that alternative targets of nimodipine beyond L-type voltage-gated calcium channels (VGCCs) may exist on or in certain cell types. Previous studies linked nimodipine-mediated effects to the activation of protein kinase B (*Akt*) and cyclic adenosine monophosphate response element-binding protein (*Creb*) signalling pathways, yet these genes were not altered in our RNA sequencing data set [28]. Owing to its hydrophobic character, it is conceivable that nimodipine enters the cytoplasm, where it binds to an intracellular target. It will be the task of future studies to identify nimodipine’s mode of action at the molecular level, which could help to unravel key pathways of myelination and neuroprotection, which, in turn, could be used towards neuroprotective drug design. Along these lines, further investigation as to the effects of nimodipine on CNS-resident cells other than oligodendrocytes seems desirable.

Overall, our results raise hope that nimodipine may be a potential therapeutic option for MS, a disease defined by demyelination and the death of oligodendrocytes [7]. Nimodipine is a well-characterised and well-tolerated drug with a favourable safety profile, supported by decades of clinical experience. The only major safety concern associated with nimodipine is the occurrence of systemic hypotension, which is dose limiting, but could be overcome by targeted drug delivery approaches. To date, most therapeutic options for MS are based on immune modulation and suppression [9]. Targeting degenerative processes early in the disease process would be an important therapeutic addendum.

## Figures and Tables

**Figure 1 brainsci-12-00476-f001:**
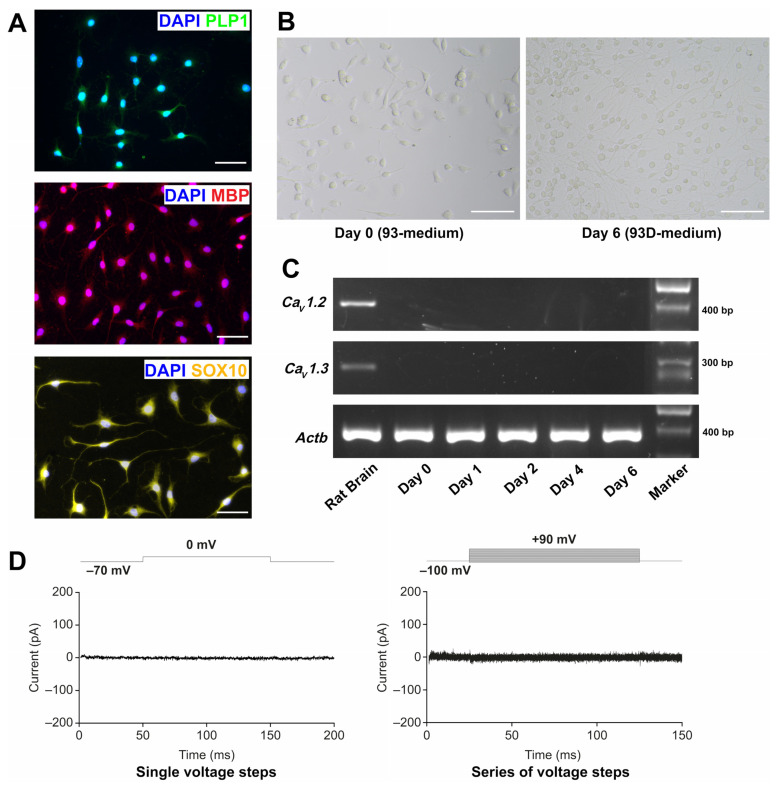
Characterisation of the OLN-93 cell line. (**A**) Immunofluorescence images of PLP1/DAPI, MBP/DAPI, and SOX10/DAPI staining of OLN-93 cells grown under normal FBS (10%) medium conditions. Results for PLP1 and MBP are representative of triplicate experiments; results for SOX10 are from one experiment. Scale bars represent 50 µm. (**B**) Light microscopic images of OLN-93 cells grown under normal FBS (10%) medium conditions (Day 0) and after 6 days in 93D-medium. Scale bars represent 100 µm. Images are representative of three independent experiments. (**C**) PCR amplification products of *Ca_v_1.2, Ca_v_1.3,* and *Actb* of OLN-93 cells grown in oligodendrocyte differentiation medium and harvested on Days 0, 1, 2, 4, and 6. Rat brain was used as a positive control. Results are representative of triplicate experiments. ‘Marker’: GenLadder 100 bp plus 1.5 kbp (Genaxxon). (**D**) Patch-clamp analysis of OLN-93 cells. Representative current response (pA) of an OLN-93 cell to a change in holding potential from −70 mV to 0 mV is shown on the left. Current response of a different cell to a series of voltage steps ranging from −100 mV to 90 mV (10 mV increments) is shown on the right. Voltage protocols are indicated above the current traces. *Actb,* actin beta; *Ca_v_*, voltage-gated L-type calcium channel gene; DAPI, 4′,6-diamidino-2-phenylindole; FCS, foetal calf serum; MBP, myelin basic protein; pA, picoamperes; PCR, polymerase chain reaction; PLP1, proteolipid protein 1; SOX10, SRY-related high mobility group-box.

**Figure 2 brainsci-12-00476-f002:**
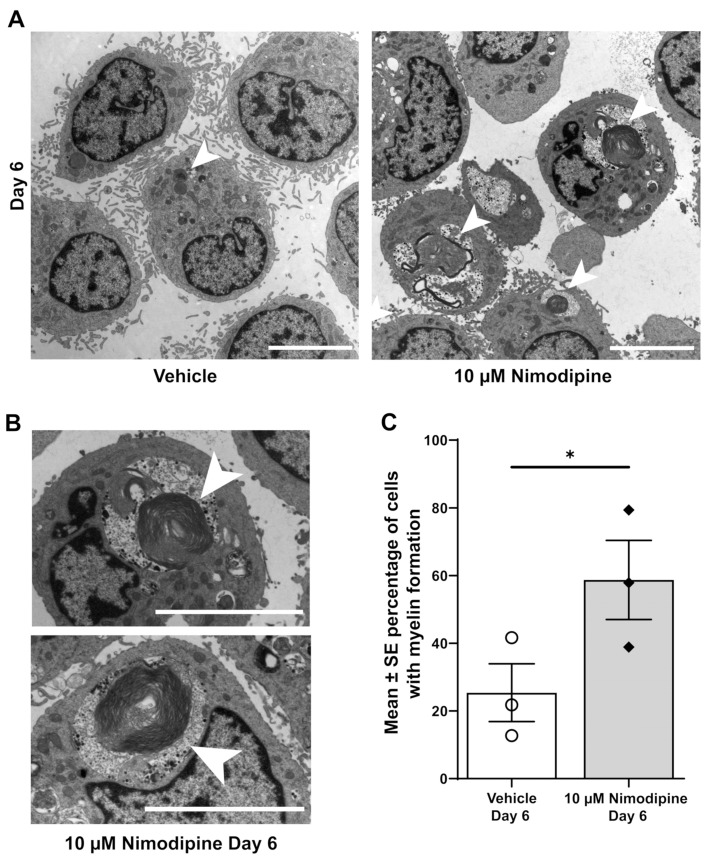
Effects of nimodipine on the ultrastructure of OLN-93 cells. (**A**) Representative transmission electron micrographs of OLN-93 cells on Day 6 treated with 10 µM nimodipine or dimethyl sulfoxide (vehicle). Representative examples of myelin formation are marked with arrows. The scale bar represents 5 µm. (**B**) Myelin formation at higher magnification marked with arrows. The scale bar represents 5 µm. (**C**) Percentage of cells with myelin formation as seen with transmission electron microscopy. Results are representative of triplicate experiments with 52–68 cells analysed per experiment. Individual data points are illustrated for each bar graph. * *p* < 0.05; SE, standard error.

**Figure 3 brainsci-12-00476-f003:**
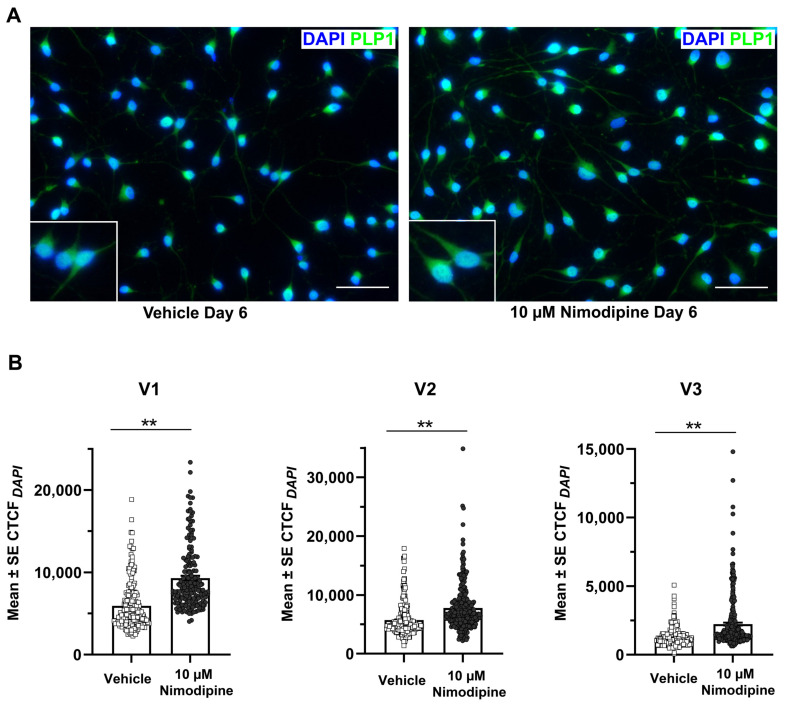
Effects of nimodipine on the PLP1 fluorescence signal. (**A**) Representative immunofluorescence images of OLN-93 cells stained for PLP1 and DAPI after incubation in oligodendrocyte differentiation medium and treatment with 10 µM nimodipine or dimethyl sulfoxide (vehicle) for 6 days. Scale bars represent 50 µm. (**B**) Quantification of the PLP1 fluorescence signal. The CTCF of PLP1 in relation to the DAPI-positive area (CTCF*_DAPI_*) was quantified in 174–438 cells each in triplicate experiments (V1–V3). Individual data points are illustrated for each bar graph. Statistical significance was determined using the Mann–Whitney *U* test. ** *p* < 0.01. CTCF, corrected total cellular fluorescence; DAPI, 4′,6-diamidino-2-phenylindole; PLP, proteolipid protein; SE, standard error.

**Figure 4 brainsci-12-00476-f004:**
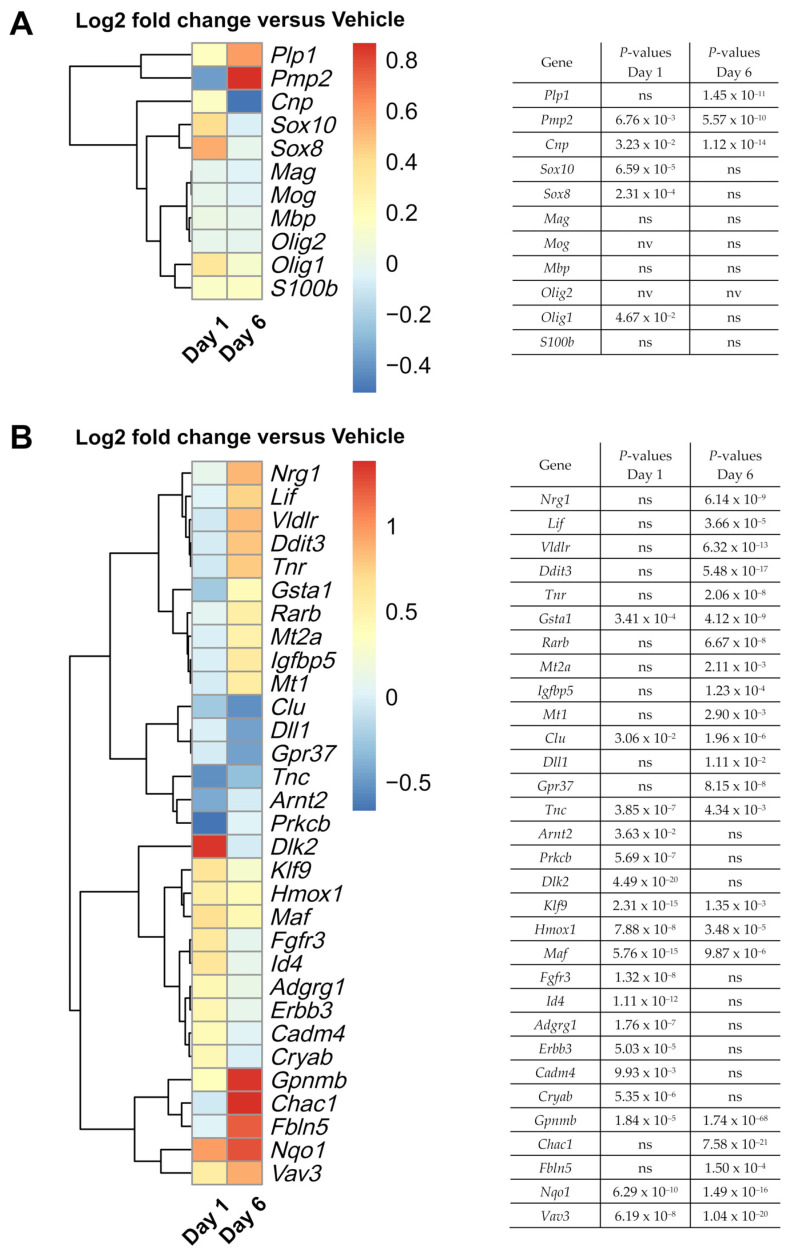
Effects of nimodipine on gene expression in OLN-93 cells, as assessed by whole-genome RNA sequencing. (**A**) Heatmap shows the effects of 10 µM nimodipine on the RNA expression of a range of common myelin-related genes on Days 1 and 6 compared with dimethyl sulfoxide (vehicle) treatment in OLN-93 cells incubated in oligodendrocyte differentiation medium. (**B**) Heatmap shows a collection of significantly differentially expressed genes in the context of oligodendrocytes, myelination, and MS, which were identified by a log2 fold change of ≤–0.4 and ≥0.4. *p*-values for (**A**) and (**B**) were corrected for multiple testing by the Benjamini–Hochberg method and are presented next to each heatmap. *Adgrg1*, adhesion G protein-coupled receptor G1; *Arnt2*, aryl hydrocarbon receptor nuclear translocator 2; *Cadm4,* cell adhesion molecule 4; *Chac1*, glutathione-specific gamma-glutamylcyclotransferase 1*; Clu,* clusterin; *Cnp*, cyclic nucleotide phosphodiesterase; *Cryab,* crystallin alpha B; *Ddit3*, DNA damage inducible transcript 3; *Dll1*, delta-like canonical Notch ligand 1; *Dlk2*, delta-like non-canonical Notch ligand 2; DMSO, dimethyl sulfoxide; *Erbb3*, ERBB2 receptor tyrosine kinase 3; *Fbln5*, fibulin 5; *Fgfr3*, fibroblast growth factor receptor 3; *Gpnmb*, glycoprotein NMB; *Gpr37*, G protein-coupled receptor 37; *Gsta1*, glutathione S-transferase alpha 1; *Hmox1,* heme oxygenase 1; *Id4*, inhibitor of DNA binding 4, helix-loop-helix protein; *Klf9*, Kruppel-like factor 9; *Lif*, leukemia inhibitory factor; *Maf*, MAF bZIP transcription factor; *Mag*, myelin-associated glycoprotein; *Mog*, myelin oligodendrocyte glycoprotein; *Mbp*, myelin basic protein gene; *Mt*, metallothionein; *Nqo1*, NAD(P)H quinone oxidoreductase 1; *Nrg1*, neuregulin 1; ns, not significant; nv, not valid; *Olig*, oligodendrocyte transcription factor; *Plp1*, proteolipid protein gene 1; *Pmp2*, peripheral myelin protein 2; *Prkcb*, protein kinase C beta; *Rarb*, retinoic acid receptor beta; RT-qPCR, real-time quantitative polymerase chain reaction; *S100b*, calcium binding protein B; SE, standard error; *Sox*, SRY-related high mobility group-box gene; *Tnc*, tenascin; *Vav3*, Vav guanine nucleotide exchange factor 3; *Vldlr*, very low-density lipoprotein receptor.

**Table 1 brainsci-12-00476-t001:** Primer sequences used in reverse transcription polymerase chain reactions.

Gene	Forward Primer	Reverse Primer
** *Actb* **	5′-AGCCTTCCTTCCTGGGTATGG-3′	5′-GCAGCTCAGTAACAGTCCGC-3′
** *Ca_v_1.2* **	5′-CGGCATCACCAACTTCGACA-3′	5′-GCATGCTCATGTTTCGGGGT-3′
** *Ca_v_1.3* **	5′-TACGTGGTGAACTCCTCGCC-3′	5′-CTTCGCTGAGTGCCACGTCT-3′

*Actb*, actin beta gene; *Ca_v_*, voltage-gated L-type calcium channel gene.

**Table 2 brainsci-12-00476-t002:** Mean (SE) fold change in *Mbp* and *Plp1* expression relative to Day 0 (baseline) in OLN-93 cells in response to nimodipine on Day 6.

Vehicle Day 6		10 µM Nimodipine Day 6	
** *Mbp* **	** *Plp1* **	** *Mbp* **	** *Plp1* **
4.026 (0.6479)	18.95 (7.631)	4.818 (0.7941)	33.51 (12.91) **

Quantification of the treatment effect with reverse transcription polymerase chain reaction using the ΔΔCT method in OLN-93 cells in oligodendrocyte differentiation medium. Cells were treated with 10 µM nimodipine or DMSO (vehicle) for 6 days. *Mbp*, myelin basic protein; *Plp*, proteolipid protein; SE, standard error. ** *p* < 0.01.

**Table 3 brainsci-12-00476-t003:** Characteristics of differentially expressed genes in nimodipine-treated cells and their role in the biology of oligodendrocytes, myelin, and multiple sclerosis.

Gene	Key Functions and Summary of Key Studies
*Maf; Nrg1; Erbb3*	The upregulation of *Maf* supports the idea of a promyelinating effect of nimodipine, as it was shown to be involved in cholesterol synthesis in myelinating Schwann cells linking *Nrg1* to cholesterol synthesis via the tyrosine kinase receptor gene *Erbb3* [60].
*Vldlr*	*Vldlr* was reported to be expressed in mature myelinating oligodendrocytes in the postnatal CNS of mice [59].
*Cryab*	*Cryab* was reported to have anti-apoptotic, anti-inflammatory and neuroprotective effects. *Cryab* knockout mice developed worse experimental autoimmune encephalomyelitis (EAE), and the administration of CRYAB ameliorated clinical symptoms in EAE mice [66]. In the peripheral nervous system (PNS), *Cryab* is a regulator of remyelination after peripheral nerve injury [58].
*Gsta1*	Glutathione transferases are involved in cell detoxification and amelioration of oxidative stress [63]. Glutathione and redox regulation are important for myelination. A deficit in glutathione was reported to impair myelin formation in patients with schizophrenia [67].
*Lif*	*Lif* is required for postnatal mouse optic nerve myelination and promotes oligodendrocyte survival [54]. In vitro and in vivo administration of LIF induced oligodendrocyte precursor cell (OPC) differentiation and myelination [52].
*Ddit 3*/*Chop*	Controversial findings indicate a pro- and anti-apoptotic role of *Ddit3*/*Chop* [68,69,70,71].
*Tnc*; *Tnr*	*Tnr* has regulatory properties in oligodendrocyte differentiation [50]. *Tnr* induces oligodendrocyte differentiation, while *Tnc* was reported to inhibit the process [51].
*Rarb*	Retinoic acid is a regulator of myelination in the PNS [72]. It is required for OPC differentiation in the postnatal mouse corpus callosum [73].
*Mt1*; *Mt2a*	*Mt2a* is the major isoform of *Mt1* and *Mt2* in the CNS. It promotes axonal regeneration in a variety of neurons [74]. In inactive multiple sclerosis (MS) lesions, *mt1* and *mt2* expression was slightly increased compared with active lesions, suggesting a role in disease remission [75]. MT2 administration ameliorated clinical symptoms in EAE mice [76].
*Igfbp5*	*Igbp5* was reported to be involved in rat brain recovery after transient hypoxic-ischaemic injury and could play an important role in oligodendrocyte regeneration [77].
*Clu*	CLU is elevated in the cerebrospinal fluid of patients with MS and may play a role as a neuro-inflammatory mediator [78,79].
*Chac1*; *Dlk2*; *Dll1*	*Chac1* and *Dlk2* play a role in antagonising the common signalling pathway *Notch* [80,81], while *Dll1* is a *Notch*-ligand [82]. *Notch1* inhibition accelerated remyelination in cuprizone-treated mice [82], whereas Notch receptor activation inhibited oligodendrocyte differentiation [83]. In EAE mice, *Notch* signalling inhibition ameliorated clinical symptoms and enhanced remyelination [84]. In a model of ischaemic injury, *Chac1* had neuroprotective properties by antagonising *Notch1* maturation [80].
*Gpr37*	*Gpr37* is a negative regulator of myelination and oligodendrocyte differentiation [53].
*Arnt2*	*Arnt2* expression decreased with the maturation of the OPC cell line Oli-neu. *Arnt2* knockdown increased the number of myelinating oligodendrocytes [85].
*Prkcb*	Protein kinase C (*Prkc*) was reported to have a proliferative effect on immature oligodendrocytes while at the same time having an inhibitory effect on oligodendrocyte differentiation. In mature oligodendrocytes, *Prkc* increased process extension and myelin formation [86].
*Hmox1*	*Hmox1* knockout led to increased demyelination, paralysis, and mortality in EAE mice. Increased *Hmox1* expression suppressed autoimmune neuroinflammation [56].
*Klf9*	*Klf9* is required for normal myelin regeneration in cuprizone-treated mice [55].
*Fgfr3*	Regulator of oligodendrocyte development. Expression of *Fgfr3* was increased when late oligodendrocyte progenitors entered terminal differentiation. *Fgfr3-*deficient mice showed a reduced number of differentiated oligodendrocytes and delayed myelination [87].
*Id4*	Overexpression of *Id4* may inhibit oligodendrocyte differentiation, yet it seems to differentially regulate the expression of myelin genes as decreased levels of P*lp1* were found in *Id4*-null mice [88].
*Adgrg1 (Gpr56)*	Regulator of oligodendrocyte development. Loss of *Gpr56* leads to hypomyelination. Transient overexpression leads to OPC proliferation. *Gpr56* was relevant for remyelination in cuprizone-treated mice [89].
*Cadm4*	Important for axoglial adhesion and correct placement of the myelin sheath [90,91].
*Gpnmb*	*Gpnmb* may play a role in neuroprotection as it led to prolonged survival in a mouse model of amyotrophic lateral sclerosis and reduced infarct volume in an ischaemic injury mouse model [64,65].
*Fbln5*	It plays an important role in Schwann cell development and could restore myelination in a zebrafish model of Charcot-Marie-Tooth type 1 disease (CMT1), a demyelinating disease of the PNS [61]. A subtype of autosomal CMT1 is associated with a mutation in the *Fbln5* gene [92].
*Nqo1*	An antioxidant protein that may be an indicator of oxidative stress, but it may also be important for myelination [62]. Genetic polymorphisms of the detoxification-associated genes *Nqo1* and *Gstp1 (*glutathione S-transferase P1*)* might be relevant for susceptibility and clinical presentation in patients with MS [93].
*Vav3*	A regulator of myelination and oligodendrocyte maturation. In the *Vav3* knockout of cerebellar slice cultures, remyelination was impaired as strongly as in cuprizone-treated *Vav3* knockout mice [57].

*Adgrg1*, adhesion G protein-coupled receptor G1; *Arnt2*, aryl hydrocarbon receptor nuclear translocator 2; *Cadm4*, cell adhesion molecule 4; *Chac1*, glutathione-specific gamma-glutamylcyclotransferase 1*; Clu,* clusterin; *Cnp*, cyclic nucleotide phosphodiesterase; *Cryab,* crystallin alpha B; *Ddit3*, DNA damage inducible transcript 3; *Dll1*, delta-like canonical Notch ligand 1; *Dlk2*, delta-like non-canonical Notch ligand 2; *EAE*, experimental autoimmune encephalitis; *Erbb3*, ERBB2 receptor tyrosine kinase 3; *Fbln5*, fibulin 5; *Fgfr3*, fibroblast growth factor receptor 3; *Gpnmb*, glycoprotein NMB; *Gpr37*, G protein-coupled receptor 37; *Gsta1*, glutathione S-transferase alpha 1; *Hmox1,* heme oxygenase 1; *Id4*, inhibitor of DNA binding 4, helix-loop-helix protein; *Klf9*, Kruppel-like factor 9; *Lif*, leukemia inhibitory factor; *Maf*, MAF bZIP transcription factor; *Mt*, metallothionein; *Nqo1*, NAD(P)H quinone oxidoreductase 1; *Nrg1*, neuregulin 1; ns, not significant; nv, not valid; *Prkcb*, protein kinase C beta; *Rarb*, retinoic Acid Receptor beta; *Tnc*, tenascin; *Vav3*, Vav guanine nucleotide exchange factor 3; *Vldlr*, very low-density lipoprotein receptor.

## Data Availability

The data sets used and analysed during the current study are available from the corresponding author upon reasonable request.

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
