# Peer review of "Nimodipine Exerts Beneficial Effects on the Rat Oligodendrocyte Cell Line OLN-93"

_brainsci, 2022, doi:10.3390/brainsci12040476_

Round 1

Reviewer 1 Report

Nimodipine belongs to the group of dihydropyridines, a well-characterised and approved drug class for the treatment of subarachnoid haemorrhage. Preclinical studies have demonstrated various beneficial effects of nimodipine on the CNS, among which improving cognitive dysfunction after surgery ameliorating cerebral ischaemia consequences in rats. In addition, nimodipine improved symptoms in experimental autoimmune encephalomyelitis (EAE) suggesting its potential efficacy in MS.  Clinical studies have shown a positive effect on vocal fold, facial motion and an improved hearing outcome after vestibular schwannoma surgery. Overall, nimodipine seems to affect several different cell types of the nervous system, including oligodendrocytes, Schwann cells, astrocytes, microglia and neurons. The current study aimed to address the question of whether nimodipine exerts direct effects on oligodendrocytes using the rat oligodendrocyte cell line OLN-93.

As the author stated the OLN-93 cell line does not express proper myelin markers during oligodendrocyte maturation therefore they restricted their aim to “ investigate the effects of the nimodipine on RNA expression of Cav1.2 and Cav1.3 during the OLN-93 process of differentiation”

Thy found that the “RNA expression of both Cav1.2 and Cav1.3 was not detectable in OLN-93 cells”. Despite this they carried out voltage-clamp recordings from cultured OLN-93 cells in the whole-cell configuration of the patch-clamp technique at a holding potential of –70 mV. The result showed that  “neither a single voltage step to 0 mV nor a series of increasing voltage steps induced a measurable current (n = 5) (Figure 1D).”

They next examined the effects of Nimodipine on expression of the myelin genes Plp1 and Mbp. “Day 6 Plp1 expression showed a significant increase,” while “Mbp expression on Day 6 was limited”. They also identify Myelin-like structures in OLN-93 cells using transmission electron microscopy, which have a peculiar appearance

All together these results suggest that the expression of myelin genes in OLN-93 is deregulated and therefore the differentiation of OLN-93 is not representative of typical oligodendrocyte maturation (previously reported by many authors among which Zuchero et al., 2015 and  Nawaz et al., 2015)

The Effects of nimodipine on gene expression in OLN-93 cells by whole-genome RNA sequencing was assessed.

All together this study show that OLN-93 do not develop as typical pre-OL into mature myelinating OLs. The interpretation of the promyelinating effect of nimodipine in this cell line have to be therefore taken with caution when the intention is to demonstrate its activity on remyelination and oligodendrocyte differentiation.

Author Response

We would like to thank the reviewer for the detailed comment. We now discuss the lack of typical differentiation patterns in OLN-93 cells more critically on page 14 of the revised manuscript. In addition, we have deleted the sentence on the pro-myelinating effect of the drug on the same page.  

Reviewer 2 Report

This study by Kuerten group describes the effects of nimodipine on OLN-93 cell line. As a group focused on MS, this is a relevant study to this group, and it is well written, designed and presented nicely. 

The only comments I have are the following:

  • In the introduction, more information should be given on nimodipine: more info on tests used, safety, side effects, mode of operation, etc.
  • The in vitro characterization disregard other cells' effect and is limited in its implication as to the drug's effects on OLs in vivo. This should be emphasized in the introduction and discussion. 
  • Catalogue number and more information on the commercial drug (purity?) should be given. 
  • It is unclear what the yellow signal in fig.1b represents? please clarify. 
  • Information on the currents given in fig.1d should be given on the figure itself. Also, in fig.1 legend, a typo was found, not having "(D)" to start the panel D explanation (instead you have "(C)". 

Author Response

We would like to thank the reviewer for the very positive and helpful comments.

Following the suggestions, we have added more information on nimodipine in the introduction on page 2 and in the discussion on page 15 of the revised manuscript. In addition, we emphasize more clearly that our study only has implications for nimodipine’s effects on oligodendrocytes (page 15). We have added the technical information regarding nimodipine on page 3 of the revised manuscript and we have replaced the images in Figure 1B. In addition, we have added the requested information to Figure 1D and we have corrected the figure legend.